# Study on the Influence of the Ball Material on Abrasive Particles’ Dynamics in Ball-Cratering Thin Coatings Wear Tests

**DOI:** 10.3390/ma14030668

**Published:** 2021-02-01

**Authors:** Gustavo Pinto, Andresa Baptista, Francisco Silva, Jacobo Porteiro, José Míguez, Ricardo Alexandre

**Affiliations:** 1ISEP—School of Engineering, Polytechnic of Porto, Rua Dr. António Bernardino de Almeida 431, 4200-072 Porto, Portugal; gflp@isep.ipp.pt (G.P.); absa@isep.ipp.pt (A.B.); 2INEGI—Instituto de Ciência e Inovação em Engenharia Mecânica e Engenharia Industrial, Rua Dr. Roberto Frias 400, 4200-465 Porto, Portugal; 3UVIGO—School of Industrial Engineering, University of Vigo, Lagoas, Marcosende, 36310-E Vigo, Spain; porteiro@uvigo.es (J.P.); jmiguez@uvigo.es (J.M.); 4TEandM—Tecnologia e Engenharia de Materiais, S.A., Parque Industrial de Taveiro, Taveiro, 3045-504 Coimbra, Portugal; ricardo@teandm.pt

**Keywords:** micro-abrasion, ball-cratering, PVD coating, TiN, abrasive particles, diamond, alumina, rolling, grooving

## Abstract

Micro-abrasion remains a test configuration hugely used, mainly for thin coatings. Several studies have been carried out investigating the parameters around this configuration. Recently, a new study was launched studying the behavior of different ball materials in abrasive particles’ dynamics in the contact area. This study intends to extend that study, investigating new ball materials never used so far in this test configuration. Thus, commercial balls of American Iron and Steel Institute (AISI) 52100 steel, Stainless Steel (SS) (AISI) 304 steel and Polytetrafluoroethylene (PTFE) were used under different test conditions and abrasive particles, using always the same coating for reference. Craters generated on the coated samples’ surface and tracks on the balls’ surface were carefully observed by Scanning Electron Microscopy (SEM) and 3D microscopy in order to understand the abrasive particles’ dynamics. As a softer material, more abrasive particles were entrapped on the PTFE ball’s surface, generating grooving wear on the samples. SS AISI 304 balls, being softer than the abrasive particles (diamond), also allowed particle entrapment, originating from grooving wear. AISI 52100 steel balls presented particle dynamics that are already known. Thus, this study extends the knowledge already existing, allowing to better select the ball material to be used in ball-cratering tests.

## 1. Introduction

Rutherford and Hutchings [1,2] were the pioneers in pointing out the ball-cratering test as a promising way to characterize thin hard coatings regarding abrasive wear, developing the corresponding theory and drawing up possible applications. Indeed, this test configuration presents the following advantages: (a) particularly useful for testing thin coatings or small volumes of material; (b) equipment is small, economical and easy to use; and (c) high versatility. Later, some researchers led by Gee et al. [3] outlined the rules that could lead to the establishment of a standard and interpretation of results, dissecting the test principles and some possible configurations, as well as the test parameters and uncertainties around the measurement of the craters. This work turned out to be the precursor to another one that appeared two years later [4], where the main goal was to analyze the accuracy, repeatability and reproducibility of the test involving fourteen laboratories, considering two measuring methods: profilometry and optical microscopy. It was concluded that optical microscopy provided a greater reproducibility of results, with the two differing by only 2% among the laboratories involved. Later, Schiffmann et al. [5] pointed out several problems with the measurement of the crater by optical means, due to overestimation of the crater diameter when perforation occurs, concluding that profilometry is the most suitable way to quantify the wear in micro-abrasion tests. However, researchers’ preference for optical measurement has been obvious. Another group of researchers [6] worked in order to establish a set of guidelines for ball-cratering tests, attempting to avoid errors in crater measurement. Gee [7] recommended the use of a scanner connected to a personal computer, recognizing this way the crater borders, concluding that the associated error readings of the scanner system relative to optical microscopy did not compromise the wear evaluation, not limiting the reading accuracy. However, this technique does not seem to have gathered followers, since there are no significant works using it. Further, Leroy et al. [8] reported some problems related to the ball-cratering test. In fact, it was possible to conclude that the concentration of abrasive particles in the slurry used, as well as the size of the particles themselves, has a direct influence on the wear mechanism developed in the micro-abrasion process, which normally assumes one of two variants: two-body abrasion, also called grooving, and three-body abrasion, commonly referred to as rolling. Moreover, when the coating to be tested has a very high hardness, the wear mechanism observed is very close to two-body abrasion due to the high wear resistance offered by the coating annulus, after the coating has been perforated.

Due to the uncertainties previously referred to, researchers started analyzing the influence of the test parameters. Stachowiak et al. [9] studied the micro-abrasion phenomena when using large particles of different abrasive materials: glass beads, silica sand, crushed quartz and alumina. The authors found in that work that the ball’s roughness grew consistently throughout the tests, verifying as well that increased roughness directly impacted the measured wear rate, as well as the wear mechanisms developed. It was also found that the surface roughness of the balls grew even faster when the abrasive particles had more angular shapes. Based on these observations, it would be impossible to expect the wear rate to show a linear behavior. In addition, this lack of linearity is further affected by the increased capacity of abrasive particles to enter the crater, as wear makes the crater grow. Thus, it was possible to verify that as the ball’s roughness and crater’s diameter increased, the dominant wear mechanism changed from three-body abrasion (rolling) to two-body abrasion (grooving). However, even before Stachowiak’s study, Shipway [10] drew attention to the need for a careful interpretation of the wear mechanisms in micro-abrasion tests, noting that any change in the test conditions led to inevitable implications for the particles’ dynamics in the micro-abrasion process. Since the applied load can determine different shapes of the particles to move across the contact zone, Shipway [10] settled a model that made it possible to predict the dynamics of particles across the contact depending on the normal load to which they were subjected, as well as to predict how these abrasive particles transmitted the load. This author also drew attention to the importance of the abrasive particles’ shape, which has a strong influence on their movement through the contact zone. Experimental validation studies of the model showed that the laboratory results obtained were in good agreement with the predictions elaborated by the model. Cozza et al. [11] also investigated the behavior of the abrasive particles within the contact area, concluding that by keeping the load and the pressure constant, a very low variation in the friction coefficient is attained. However, a later work of the same authors [12] showed that the degree of micro-rolling along the grooves tends to decrease as the normal load is increased. Moreover, when the normal load is increased, the degree of dispersion of particles through contact tends to decrease. Other researchers [13] also concluded that it is impossible to compare micro-abrasion results with tests carried out with different abrasives or different shapes of particles. Indeed, some authors have tested a wide range of normal loads in diamond coating wear characterization, concluding that 0.3 N of normal load, 75 rpm of ball rotation and 0.5 g/mL of 4–6 µm SiC particles abrasive concentration are the ideal test conditions, promoting rolling abrasive wear in a stable way. Micro-abrasion was also used to find potential wear transitions in multilayered coating systems [14], making it possible to conclude that while monolayered coatings show clearly grooving abrasive wear, the multilayered coatings present a mixed behavior of grooving and rolling abrasive wear. The influence of the abrasive particles’ size was also investigated [15] using three different sizes of SiC (silicon carbide) abrasive particles and maintaining the other test parameters constant. It was concluded that particles provided with larger size (F800) had increased difficulty to enter the contact area and carry the load from the ball to the coated sample, but when they enter the contact, they act as a third body, rolling across the contact in a random way. On the other hand, smaller particles (F1000 and F1200) essentially promoted grooving. Another study performed by Silva et al. [16] investigated three different abrasive materials with similar particle sizes, concluding that diamond particles produced more accurate circular craters than alumina or silicon carbide abrasive particles. Further, the ball’s material has been investigated, trying to study the friction coefficient behavior [17]. It was concluded that harder balls induce a greater friction coefficient, while softer ones tend to entangle the abrasive particles on their surface, showing rolling wear behavior in an earlier stage and evolving into grooving during the test. The effect of the rotation speed of the ball in these tests has also been investigated [18], concluding that a higher ball speed induces a decrease in the wear coefficient, entrapping more abrasive particles on the ball’s surface. The influence of the abrasive concentration and the normal load has also been investigated by Cozza [19]. That study allowed concluding that there is no direct relation between the abrasive concentration or normal load and the friction coefficient. Another study [20] was aimed at understanding the influence of samples’ hardness on the trajectory of the abrasive particles throughout the contact area. It was concluded that a lower number of abrasive particles cross the contact carrying the normal load when they are larger and with a more irregular shape. In these conditions, the stress promoted by each particle is higher, inducing a lateral fracture and a greater wear rate. Moreover, when the surface of the ball is rougher, the dragging effect of the abrasive particles is increased in the contact area. The same study also concluded that smother samples’ surface is induced for a broader movement of the abrasive particles, increasing the wear rate. Similar experiments were performed by Baig et al. [21] using different abrasive particle sizes and shapes, varying their concentration and normal load applied as well in the contact, corroborating observations previously made that the particles’ concentration and their size have a clear influence on the wear rate, which increases with the particles’ size and augmented concentration. The strong negative influence of the angular shape of the particles on the wear rate was also corroborated.

Several coatings have been tested using a ball-cratering configuration, both at room temperature and at high temperature, as performed by Allsopp and Hutchings [22], exploring the wear behavior at room temperature and at 350 °C of AlTiN and TiCN coatings deposited by Physical Vapor Deposition (PVD). However, at high-temperature conditions, abrasive particles were used in dry conditions. Through that work, it was possible to conclude that wear behavior is not the same at room temperature and high temperature due to different scratch coating conditions and thus the results cannot be correlated when different temperature conditions are used. In that work, just the AlTiN showed similar wear behavior at both temperatures. Stack and Mathew [23] drew maps correlating the sliding distance and normal load with the wear level regarding WC/Co coatings, concluding that grooving and rolling abrasive wear domains are not close to each other. Similar studies were carried out by Rodríguez-Castro et al. [24] testing a boride coating, finding the conditions able to change the abrasive wear behavior from grooving to rolling. The abrasiveness of extremely hard coatings such as Chemical Vapor Deposition (CVD) diamond was also tested by Silva et al. [25], but, in this case, the authors found that too high normal loads should not be used when using thick intermediate layers of soft materials. Testing nanocrystalline diamond (NCD) coatings synthetized by Hot-Filament Chemical Vapor Deposition (HFCVD) and Microwave Plasma Chemical Vapor Deposition (MPCVD) on substrates of Si_3_N_4_, Silva et al. [26] also found that HFCVD coatings revealed better wear resistance than MPCVD ones, despite poorer adhesion to the substrate of the former ones. Testing TiN and TiC coatings using a ball-cratering configuration, Cozza [27] found that the friction coefficient for both coatings spans between 0.4 and 0.9, confirming that the higher the hardness presented by the coatings, the lower the wear rate. Silva et al. [28] investigated TiAlSiN coatings’ wear behavior, concluding that a small addition of silicon, such as 5% (wt.), does not significantly increase the abrasive wear performance of the coatings, comparing with the results previously obtained using TiAlN coatings in the same conditions. However, also using ball-cratering tests, Martinho et al. [29] found that PVD TiAlCrSiN coatings offered 50% more abrasion wear resistance than uncoated heat-treated steel usually used in molds for injection of glass fiber-reinforced plastics (GFRP). Further, using a ball-cratering test configuration to test multilayered CrN/CrCN/DLC coatings with a view to their use in GFRP injection molds and comparing these results with others obtained in practical terms regarding an insert put into the mold, Silva et al. [30] reported unreliable results because micro-abrasion showed moderate improvements, but the inserts showed an excellent performance in terms of wear abrasion.

In this work, it was intended to investigate the influence of the material type of the balls in the wear mode, for two different abrasives, keeping the same particle size. Regarding the characteristics of each material selected for the balls, dynamics of the particles were studied, characterizing the wear behavior offered induced by each pair of ball material/abrasive particle material, keeping constant the test conditions and the coating.

## 2. Materials and Methods

### 2.1. Material

#### 2.1.1. Substrate Material and Geometry Characterization

For conducting the micro-abrasion tests, samples were prepared that underwent a PVD deposition process, with the following dimensions: 28 mm × 25 mm and 4 mm thick. The substrate used was uddeholm calmax, DIN: X210Cr12, AISI D3, W.NR.: 1.2080 (F. Ramada S.A, Porto, Portugal), in the annealed state, with the following characteristics: hardness 248 HB and Young modulus of 210 GPa. The chemical composition of calmax (chromium molybdenum-vanadium alloyed steel) steel supplied by the supplier (F. Ramada S.A, Porto, Portugal) is the following: C—1.95; Si—0.35; Mn—0.6; Cr—11.5 (wt%). In the PVD deposition process, the samples were subjected to a three-stage heat treatment, annealing for stress relief, hardening and tempering according to the supplier’s technical data sheet, obtaining a final mean hardness of 63 ± 0.5 HRC. After heat treatment, the samples’ surface was cleaned by ultrasound, ground with sandpaper of different grains and diamond polished until reaching a surface roughness Ra of 0.0263 ± 0.002 µm, using a Mahr Perthometer M2 (MAHR, GmbH., Göttingen, Germany), carrying out five measurements in two orthogonal directions and calculating the average value and corresponding standard deviation.

#### 2.1.2. Ball Characterization

Micro-abrasion wear tests were carried out on a ball crater tribometer (Phoenix Technology, London, UK) to study the influence of the ball’s surface texture on the dragging of abrasive particles. For the micro-abrasion tests, balls of SS AISI 304 (28 HRC ± 2 HRC) (Rolveda Lda., S. Mamede de Infesta, Portugal) and AISI 52100 steels (63 HRC ± 3 HRC) (Rolveda Lda., Porto, Portugal) and polytetrafluoroethylene PTFE (58 ± 6 Shore D) (Rolveda Lda., S. Mamede de Infesta, Portugal), with a diameter of 25 mm, were acquired. To wash the surface of the possible protective layer of lubricant or other dirt, the balls were cleaned in an ultrasonic acetone bath for 5 min. For the tests, 2 polished SS AISI 304 steel balls, 2 polished AISI 52100 steel balls and 2 Teflon PTFE balls were used. To better understand the influence of the texture and roughness of each ball, these were observed by SEM (Field Electron and Ion Company, Hillsboro, OR, USA), Figure 1, and by non-contact 3D optical profilometry Bruker NPFLEX equipment (Bruker Optics Inc, Billerica, Boston, MA, USA). This is the focus of this work since it is intended to study the extent to which the cavity dimensions of each ball are capable of retaining or not the abrasive particles used to promote micro-abrasion on the coated surface.

#### 2.1.3. Abrasive Particles Characterization

For this work, two types of abrasives were used, MicroPolish Alumina Powder-Deagglomerated Alpha (1 µm) (Microdiamant USA, Inc., Smithfield, PA, USA) and Monocrystalline Diamond Powder MSY (1–2 µm) (Buehler, Lake Bluff, IL, USA). Since the shape of the particles has an important role in the wear process, the particles corresponding to the abrasives used, with similar granulometry, were analyzed by SEM. Figure 2a,b show their geometry.

The influence of particle distribution on the wear rate and the wear mechanisms developed during micro-abrasion tests has been already studied [21]. It was understood that the particle size should be validated by conducting a study of their distribution. Samples of the abrasives under study were collected, and these were dispersed in water at 250 rpm and the readings were taken five times.

For this study, a piece of laser diffraction equipment, the Malvern 2000 Particle Size Analyzer (Malvern Instruments Ltd, Malvern, UK), with a Hydro 2000G sample dispersion unit, was used. It should be noted that the equipment used applies mathematical models (Mie or Fraunhofer theory) to generate a particle size distribution. Thus, a result is presented based on the equivalent spherical diameter volume. One parameter that shows the range of the size distribution is span (indirect measurement of particle distribution).

The size distribution range based on volume is defined as span = (D90 − D10)/D50, which allows perceiving how far the 10% and 90% points are separated, normalized to the midpoint. In measurements of size distribution by laser diffraction, the use of parameters D90, D50 and D10 is common. The parameter D90 represents the point in the size distribution where up to 90% of the total volume of material in the sample is “contained”. D50 is the size point below which 50% of the material is contained. Likewise, D10 is the size below which 10% of the material is contained. The distribution data were collected and analyzed according to the volume weighted mean D {4, 3}. The volume moment mean (De Brouckere mean diameter) reflects the size of the particles that make up most of the sample volume being more sensitive to the presence of large particles in the size distribution. The obtained values for the analyzed abrasives are as follows:

Alumina powder D 1 µm: span = (7.945 − 3.013)/2.259 = 2.259 µm, volume weighted mean D {4, 3} = 3.885 µm, and diamond powder D 1–2 µm: span = (4.499 − 1.161)/2.350 = 1.421 µm, volume weighted mean D {4, 3} = 2.625 µm. Figure 3 shows the result of the particle size distribution for abrasives diamond powder 1–2 µm and alumina powder 1 µm.

The analysis of the granulometry, according to the weighted average of Volume D {4, 3}, allowed perceiving the dispersion of the particle size. Thus, it was concluded that the average particle size, relative to the diamond dust, is within the expected range, 1–2 µm. On the other hand, 1 µm alumina powder has an average particle size value higher than expected. However, as it is intended to evaluate two different abrasives with similar particle sizes, it can be concluded that it is possible to validate the abrasives for the tests since the abrasives guarantee a similar particle diameter for this study.

### 2.2. Methods

#### 2.2.1. Coating Deposition Process

A CemeCon CC800/9ML sputtering reactor (CemeCon AG, Würselen, Germany) was used to coat the samples. The thin TiN film about 4.5 mm thick resulted from the use of four Ti targets. In the deposition process, three gases with 99.99% purity were used: argon, nitrogen and krypton, with the following parameters: deposition time 123 min, temperature 480 °C, gas pressure 650 mPa, target power density 20 A cm^−2^ and polarization in the range of −105 to −90 V. The homogenization of the film composition resulted from the rotation speed of the sample support of 1 rpm.

#### 2.2.2. Morphology and Thickness Analysis

To measure the thickness of the film, after depositing the PVD, the samples were cut at the back with a disc saw to obtain a deep channel, then the samples were immersed in liquid nitrogen for 30 min and then carefully broken down in the cold by mechanical means. This method was followed to avoid mechanical deformation of the substrate and the coating close to the area that was broken, and this allowed an adequate evaluation of the coating thickness. Thus, a FEI Quanta 400 FEG SEM (Field Electron and Ion Company, Hillsboro, OR, USA) with an EDAX Genesis X-ray spectroscope (EDS) (FEI, Hillsboro, OR, USA) was used to measure the thickness and characterize the surface morphology.

#### 2.2.3. AFM and Profilometry, Roughness Analysis

Two methods were used to assess the surface roughness of the coated sample. In the first method, the Mahr Perthometer M2 profilometer (MAHR, GmbH., Göttingen, Germany) equipment was used, equipped with an NHT 6-100 probe, where ten measurements were made. The test was carried out according to DIN EN ISO 4288 and ASME B46 [31,32], seven segments of 0.8 mm each (cut-off) were used in all measurements. The first and last measurements were discarded due to the initial acceleration and final deceleration processes. To assess the roughness, the following parameters were considered: the arithmetic average of the surface roughness (Ra) and the maximum roughness height (Rmax) according to EN ISO 4287 [33]. The variables are expressed in values of R (Ra and Rmax) since their analysis was performed in 2D. The roughness was confirmed using an atomic force microscope (AFM), VEECO Multimode (VEECO Instruments, Ltd., Woodbury, NY, USA), as the second method. To reach the largest possible area with high precision, an area of 20 × 20 μm^2^ was analyzed and three different analyses were performed. The equipment used was equipped with a 0.007 µm radius probe and with NanoScope 6.13^®^ software (Bruker, New York, NY, USA).

To understand the morphological pattern of the balls, they were analyzed by non-contact optical profilometry on the Bruker NPFLEX equipment (Bruker Optics Inc, Billerica, Boston, MA, USA) with Vision64^®^ software (Bruker Optics Inc, Billerica, Boston, MA, USA). Three different analyses were made, taking into account an area of 450 × 600 μm^2^, of the roughness parameters Sa and Sz according to ISO 25178-2:2012 [34]. The 3D techniques provide a better understanding of the surface, so in order to characterize the surface of the balls, the following parameters were analyzed: Sz (maximum topographic surface height) and the arithmetic mean of the surface roughness, Sa. The Sa parameter allows a quick perception of the state of the surface; however, it can be deceptive since it is an average value. On the other hand, the parameter Sz corresponds to the amplitude values between the highest peak and the deepest valley of the five regions of the image, which allows a more comprehensive description of the topography.

PTFE and steel balls 304 and 52,100 were previously cleaned in an ultrasonic acetone bath for five minutes for topographic analysis.

#### 2.2.4. Adhesion Analysis

The adhesion between the TiN film and substrate was assessed by both a scratch test and a Rockwell indentation test. The scratch test allows the quantification of the normal load adhesion between the film and the steel substrate in cohesive (Lc1) and adhesive (Lc2) failure modes. A posterior optical microscopy analysis allows the identification of the local and corresponding normal load responsible for the failures at the interface. The scratch tests were carried out using a Rockwell indenter which slides at a speed of 10 mm/min and using a load range of 0 to 30 N at a load increasing rate of 10 N/min. This indenter produces a progressive scratch due to the action of the 100 µm Rockwell C diamond indenter I-119 under the imposed increasing load. This procedure was repeated six times for each sample, to increase results’ accuracy. Three tests were carried out in two orthogonal directions, to identify possible different behaviors caused by grinding and the polishing texture. The scratch tests were developed by CSM Revetest scratch tester (CSM Instruments, Peseux, Switzerland), provided with an acoustic emission detector, according to the BS EN ISO 20502:2016 standard [35].

Rockwell indentation is used by many researchers as a way of ascertaining the adhesion of PVD or CVD films to the substrate. This analysis is based on the behavior of the film through an indentation performed with a conical diamond Rockwell-type indenter. The indentations were carried out in an EMCO M4U Universal Hardness Tester (model M4U, EMCO-TEST Prüfmaschinen GmbH, Kellau, Kuchl, Austria) using a load of 150 kgf and were observed using an optical microscope (OM), OLYMPUS BX51M (Olympus, Tokyo, Japan), with magnifications of 100×. Moreover, the failure modes were assessed according to the VDI 3198:1991 standard [36], allowing for an adhesion classification regarding the look of the indentation border.

#### 2.2.5. XRD Analysis

The structural characterization of the samples was focused only on the TiN film, applying the X-ray diffraction (XRD) technique, using a Rigaku SmartLabSe diffractometer (Rigaku, Tokyo, Japan). The technical characteristics and methods of the equipment used are as follows: equipment voltage: 40 kV; current: 30 mA; radiation: Cu Kα with a wavelength of 1.5406 Å; time: approximately 2 h, step size [2θ]: 0.01° and scan step time: 1.0 s, method: top of smoothed peak.

#### 2.2.6. Micro-Hardness Assessment

To analyze the hardness of the TiN film, a CSM Nanoindenter provided with a Berkovich diamond indenter was used (CSM Instruments, Peseux, Switzerland), properly calibrated for the effect. In order to improve the accuracy of the measurement, ten indentations were made. The Oliver and Pharr method [37,38] was used. The equipment used to perform the measurements was an NHTX S/N: 01-02934 (CSM Instruments, Peseux, Switzerland) using the following settings: acquisition rate: 10 Hz, max load: 150 mN, loading rate: 200 mN/min, unloading rate: 200 mN/min and dwell time of 30 s.

#### 2.2.7. Micro-Abrasion Test

The micro-abrasion equipment Plint TE-66 (Phoenix Technology, London, UK) was used in the wear tests. Two different pastes were prepared as abrasives, 1–2 µm diamond powder and 1 µm alumina powder. For each paste, 35.4 g of abrasive powder was prepared in 100 mL of distilled water.

The rotation speed of each type of ball: SS AISI 304 steel, AISI 52100 steel and PTFE, was 80 rpm, which corresponds to a speed of 0.105 m/s, and two calibrated loads of 0.2 and 0.5 N were used. For each load, five tests were carried out with a duration of 500 cycles, which corresponds to a length of 39.27 m. Figure 4a shows an image of the ball-cratering tribometer used in this work to perform the micro-abrasion tests and Figure 4b shows, in detail, a schematic diagram of the abrasive flow into and around the contact area between the ball and the samples. After testing, micro-abrasion craters were observed and measured by SEM and OM using the Olympus (BX51M).

After measuring the diameter of the craters via SEM for each of the test conditions, the volume of material removed (Equation (1)) as well as the wear rate, following the Archard expression (Equation (2)), was calculated.
(1)$$V=\pi \frac{d^{4}}{64\times r}$$

The volume of material removed V is expressed in mm^3^, d represents the diameter of the crater and r represents the radius of the ball used in the test, both expressed in mm.
(2)$$K=\frac{V}{s\times F}$$

The wear coefficient K is expressed in mm^3^/Nm, V represents the volume of material removed, expressed in mm^3^, s represents the sliding distance, expressed in m, and F represents the applied normal load, expressed in N [39].

## 3. Results and Discussion

### 3.1. Coating’s Morphology and Thickness

As described in Section 2.2.2, a previous characterization of the coating’s morphology and thickness by SEM was carried out. The surface morphology has a homogeneous aspect as can be confirmed in the image revealed by SEM, Figure 5a. Figure 5b allows a sectional view of the TiN coating, and this cross-section shows its entire columnar structure from the substrate to the coating thickness of approximately 4.60 µm. Of note, the measurement detail of the first column deposited at 0.875 µm results from the different levels of polarization used in the deposition process: −105 V in a first phase to improve adhesion, and −90 V in a second phase for the remaining deposition film. The EDS spectra in Figure 5c show, as expected, the presence of elements such as titanium and nitrogen, composing the desired titanium nitride (TiN) coating. The results of these images allow us to conclude that the film has a good density and shows no signs of trapped gases in the observations of the coating cross-section.

### 3.2. Roughness Results

Two different methods were used to assess the surface roughness of the sample: profilometry and AFM. Profilometry, as preliminary evaluation, had as results on the coating surface an average surface roughness value Ra = 0.029 ± 0.002 µm and a maximum surface roughness value Rmax = 0.365 ± 0.011 µm. Later, the roughness was confirmed by AFM. This method allowed, as expected, an evaluation with greater accuracy across the analyzed area, scanning the topography of each coated sample surface, as can be seen in Figure 6. These scans allow observing once again that the surface of the coating is smooth and uniform. At the same time, roughness parameters were collected, such as the arithmetic average roughness of the surface (Ra = 0.0263 ± 0.002 µm) and maximum roughness (Rmax = 0.337 ± 0.008 µm), allowing for comparison with the values obtained by the first method. The values presented are higher than those indicated by [9,16]; however, they are lower than those presented in [12,17,25,40]. The value of Ra is very similar to that presented in [40]. This analysis allows concluding that the roughness does not compromise, in this case, the wear behavior because it is in line with the surface roughness of samples used for similar wear tests.

The topography of the ball’s surface, in the state as supplied, was investigated by non-contact (optical) profilometry (see Figure 7). It was not possible to collect images of the PTFE ball surface, given the nature of its material, preventing its observation in the same way as for other materials used in this work. The ball surfaces show a protruding surface, with Sa values of 0.015 ± 0.001 µm and Sz values of 1.788 ± 0.023 µm (Figure 7a) for the 52,100 ball, and Sa values of 0.134 ± 0.005 µm and Sz values of 7.451 ± 0.098 µm for the 304 ball (Figure 7b).

### 3.3. Adhesion Evaluation

Scratch tests were performed to evaluate the adhesion of the film to the substrate. Thus, some indentations were performed according to the VDI 3198:1991 standard [36] to get qualitative results about the adhesion between the substrate and the coating. Indentations were carried out at 150 kgf on each coated sample, and an example of the indentations produced can be seen in detail in Figure 8. Regarding the different situations exposed in the above referred standard, the cracks identified in the indentation edge can be classified as HF1, similar to [28]. Thus, the adhesion can be considered as acceptable, but it will be confirmed by scratch tests in a quantitative way.

Figure 9 depicts one of the results obtained by scratch tests, showing the acoustic emission signal corresponding to the evolution of the indenter along the scratch. It is possible to observe that there is no acoustic emission signal before 11.2 N. This value corresponds to the first event in terms of a cohesive failure (Lc1) and an adhesive failure (Lc2) happening around 12.5 N. The cohesive failure value is in line with other works [30,40], but the adhesive failure occurs at lower values than previously reported [30]. However, these values are clearly lower than those pointed out in [22] for AlTiN coatings, where values between 46 and 71 N regarding tests carried out at room temperature were reported. Comparing these results with the indentation tests previously performed, it is possible to state that adhesion is high enough to ensure a good wear behavior in terms of tribological tests.

### 3.4. Coating’s Structure

The X-ray diffraction (XRD) pattern of the thin TiN-coated samples prepared in this work is shown in Figure 10. The XRD pattern of the deposited films presents peaks corresponding to Fe and TiN that are the most prevalent. The first ones correspond to the substrate interaction. Regarding the TiN peaks, four were detected: two weaker peaks located at 2θ = 73.92°, corresponding to TiN (222), and 2θ = 36.43°, corresponding to TiN peak (111), and higher TiN peaks located at 2θ = 42.52° and 2θ = 61.57°, corresponding to TiN (200) and TiN (220), respectively. These peaks and structure agree with other investigations previously performed, where TiN peaks in the (111), (200) and (220) directions are quite common [41,42]. The TiN peak corresponding to the (222) direction is less common to be identified, but there are other peaks that are sometimes identified, such as (311), although with low intensity [41].

### 3.5. Micro-Hardness

To avoid the influence of the substrate in the hardness measurements, it was necessary to reach very low indentation depths with high resolution. The selection of a very low load and a series of indentations allowed reaching values able to ensure very reduced uncertainty. The following results were achieved: Young’s modulus = 336.53 ± 18.27 GPa and hardness = 27.30 ± 1.84 GPa. An example of the “load–displacement” curves obtained in this work can be seen in Figure 11. Both hardness and Young’s modulus are about 20% lower than the values referred to by [41] and about 10% lower than [42], also in both cases. However, considering [43], the hardness obtained in this work is about 25% lower, but the Young’s modulus is 10% higher.

Figure 11 shows the “load–displacement” curves with an intermediate creep phase, resulting from a micro-hardness test. The “load–displacement” curve shows that the indentation depth reached a little bit less than 30% of the coating thickness, showing that some substrate influence could be affecting the obtained values, lowering the hardness value, according to the discussion above. Indeed, it is usual to select indentations’ depth at lower than 10% of the coating thickness [29,30]. This also reveals a wrong load selection of the load for this type of assessment. Following the curve, it presents a slope up to about 400 nm, becoming more accentuated above 500 nm. This can be a consequence of the influence that the substrate is having on the hardness reading. Even so, the inflection of the curve is very small, which means that the substrate did not have a major influence on the presented hardness value.

### 3.6. Micro-Abrasion Analysis

After the micro-abrasion test, all samples were subjected to SEM analysis to evaluate all craters. Table 1 shows the results of the craters’ average diameter, the wear volume induced by the abrasive particles, the wear coefficient and the respective standard deviations.

The volume removed from the film was calculated to establish the wear rate of the coating, taking into account the final diameter of the crater corresponding to each test. For these calculations, the average diameters of the five tests carried out under the same conditions were considered.

Analyzing the results obtained, it was observed from the tests carried out using an alumina abrasive that the wear rate is lower than the wear values for tests performed employing diamond abrasives. This is due to the fact that the diamond abrasive material used is particularly harder than the alumina abrasive, which is in line with other studies [15,16]. There is an increase in the diameter of the craters as the load increases from 0.2 to 0.5 N; however, it should be noted that this situation is more significant when the diamond abrasive is used regardless of the type of ball material used. It was also observed that the increase in the load caused greater wear in the craters, which gave rise to larger-diameter craters, in line with that described in the micro-abrasive wear tests with a rotating ball [19]. Figure 12 shows the wear craters generated by the AISI 52100 and SS AISI 304 balls for 0.2 and 0.5 N loads using alumina abrasive particles, where it is possible to observe clear grooves generated by the grooving wear mechanism, i.e., abrasive particles attached to the ball surface, which systematically abrade the same area of the crater. It is worth to note that it was not possible to present SEM images regarding the wear craters for the PTFE ball since this material is very soft and did not induce any wear on the coating. Observing the images of the remaining craters obtained via SEM, a perforation of the coating is visible; however, the craters are not well defined. Its definition increases with increasing load and with the use of the SS AISI 304 ball, since its material is softer and therefore drags more abrasive particles. The low abrasiveness of the alumina particles is due to their lower hardness in relation to the other abrasives used in this work and the rounded shape of the particle outline. In general, irregular craters have well-defined deep grooves, and at the edges of the craters, there are many detachments of films. This phenomenon can be attributed to the concentration of alumina abrasive particles during contact, which is common to other similar phenomena already reported by other authors [16].

As can be seen in Figure 13, the perforation of the TiN coating was evident in all tests carried out with diamond abrasive particles which resulted in well-defined craters using AISI 52100 and SS AISI 304 balls. The diameter of the craters generated by the SS AISI 304 balls are larger than the AISI 52100 balls which can be attributed to the greater roughness of these balls once they drag more abrasive particles, which allows better transport of the particles, creating more wear. Regarding the close observation of the craters, a periodic grooving effect can be observed. This corresponds to a grooving effect induced by particles embedded on the balls’ surface, promoting localized abrasive wear at each rotation of the ball [11,12,19].

Diamond abrasion using PTFE balls resulted in very irregular craters with well-defined deep grooves that increase with a change in load from 0.2 to 0.5 N. It is possible to observe in detail the detachment of the coating in Figure 14a, with its EDS in Figure 14b. The craters show the presence of a mixed wear mode. The abrasive diamond particles were rolled through the contact but there is a strong contribution of the particles embedded on the balls’ surface, which promote grooving (“two-body” abrasion), highlighting the grooves’ formation.

Figure 15 and Figure 16 show the images of the craters’ surface obtained by SEM observations in the center of each crater for a better comparison. Figure 15 results from the abrasion tests promoted by a slurry containing 1 µm alumina abrasive particles, and Figure 16 results from the abrasion tests promoted by a slurry containing 1–2 µm diamond abrasive particles, both using two different normal loads (0.2 and 0.5 N) when testing with AISI 52100, SS AISI 304 and PTFE balls. The localized images allow identifying the wear of mixed abrasion where the deep grooves stand out as the load increases. The grooves are less pronounced when using SS AISI 304 balls. They also show the failure in the use of this abrasive in the wear of the coating since it has great difficulty in detaching the coating.

On the other hand, diamond particles show better abrasive behavior using the three different types of balls. Figure 16a shows an image of a smoother surface, which suffered a slight abrasive rolling wear during the micro-abrasion test when compared to Figure 16b. Both have a wear pattern showing a very smooth vertical grooving at the 0.2 N load, which is more pronounced at the 0.5 N load. In AISI 52100 balls, larger particles are embedded but in a smaller amount, which causes the most widely spaced grooving that can be seen in Figure 16b. Figure 16c,d show images of a smooth surface that also suffered abrasive rolling wear during the micro-abrasion test with very smooth grooving. The increase in load did not have much impact on its morphology. During the test using SS AISI 304 balls, the particles which come into contact turn out to be smaller but probably drag more easily and in greater quantity since their roughness is higher and a softer material is faced. As referred to previously, Figure 16e,f show images produced by PTFE balls in the same conditions of the other tests shown in Figure 16a–d, showing clear grooving phenomenon. However, it must be considered that these images have been taken into the center of the crater, where the particles carry the load with higher intensity.

### 3.7. Ball Analyses

Bearing in mind that the ball material plays a truly relevant role in the micro-abrasion process, it must be studied and deserves all the attention in this investigation. Thus, several track images generated on the balls’ surfaces under study have been taken, as can be observed in Figure 17, Figure 18, Figure 19 and Figure 20. The surface of the PFTE ball was analyzed by SEM; however, no tracks on the ball were identified. In the examined balls, a significant number of abrasive particles embedded in the surface of the balls was observed. However, looking at Figure 17 and Figure 18, the abrasive particles of alumina are less numerous on the AISI 52100 ball track when compared to the SS AISI 304 ball track. As already mentioned, their round shape and low abrasiveness contribute to the smaller amount of the embedding phenomenon on the tracks. The greater number of abrasive particles on the SS AISI 304 ball is also due to the ball’s roughness, which is higher, as well as its hardness, because SS AISI 304 is a softer material than AISI 52100, this way allowing a greater embedding of abrasive particles.

When analyzing the images collected from the balls’ tracks for the diamond abrasive, Figure 19 and Figure 20, the number of embedded abrasive particles is greater. However, analyzing Figure 19a in detail, it shows two relevant situations in the extension of the rail, one is a very marked central mark and the second one is a mark on the sides. Given the symmetry in the abrasive behavior, the center and the right side of the track were analyzed, Figure 19b, in order to study the dispersion of diamond abrasive particles. The particles in the center, Figure 19c, are smaller and in greater quantity compared to the particles on the sides, Figure 19d, which means that the larger particles tend to circulate outside the crater and are less embedded on the side of the track.

Figure 20a shows a morphologically more uniform track, made of a softer material, which allows a greater embedding of abrasive particles, which turn out to be smaller and embedded more easily and in greater quantity. Thus, as can be seen in Figure 20b, the diamond abrasive particles are smaller when compared to the abrasive particles embedded in the AISI 52100 ball track.

To better understand the aggregation of particles to the ball, a topographic analysis was made on the tracks after the tests (Figure 21 and Figure 22). As initially described, in a previous analysis carried out on the balls of AISI 52100 and SS AISI 304, it was found that balls of SS AISI 304 present the highest Sz surface roughness. Furthermore, another different behavior regarding the ball surface roughness was observed: while in the AISI 52100 steel ball, the average roughness clearly increases from the initial state to the condition after the test, from 0.015 to 0.283 µm, the SS AISI 304 balls present a smooth surface after tests, which means that a predominant rolling behavior is verified when using SS AISI 304 balls, while some localized grooving phenomenon is verified in the center of the AISI 52100 ball tracks.

After analyzing the SEM images of the tracks, it can be concluded that, although initially the average surface roughness of the AISI 52100 ball is lower after the test, its value becomes higher on the track when compared to the SS AISI 304 ball, as shown in Figure 21 and Figure 22. It should be noted that in Figure 21a, there is a localized wear in the middle of the track, while in Figure 21b, there is a more generalized wear. Indeed, what happened was that AISI 52100 ball was harder and the particles concentrated essentially in the center of the track, with small marks caused by rolling on the outermost parts of the track, as can be seen in Figure 21a or Figure 22a,b. In the case of the SS AISI 304 ball, which was softer, the wear was not concentrated on a central track but tended to wear the ball itself in a more homogeneous way where particles created smooth grooves as a whole. Looking at the images in Figure 21b or Figure 22c,d, these grooves are caused by particles that meanwhile were embedded at the crater’s edge. Taking into consideration the values presented in Figure 21 and Figure 22, the average roughness induced in the AISI 52100 ball is similar to the one induced in the SS AISI 304 ball.

This analysis also shows that in both abrasives, the response to roughness is the same, which means that the behavior does not depend on the abrasive material but essentially depends on the material of the ball, i.e., balls of AISI 52100 or SS AISI 304 are governing the wear behavior, independently of the abrasive used (alumina and diamond), showing that SS AISI 304 balls are more suitable for this kind of wear test configuration. In fact, the wear mechanisms referred to above are indirect observations because what is happening in the crater results from the particles’ dynamics and the way they are entrapped on the balls’ surface. Thus, this helps to understand the wear mechanisms developed but cannot be considered as a perfectly reliable observation.

## 4. Concluding Remarks

Rotary ball micro-abrasion has been a topic that has been consistently studied for the past two decades. The influence of several factors and parameters involved in this type of wear test has been studied, but only a recent study addressed the behavior of different ball materials in the dynamics of abrasive particles in the contact area. However, that study [17] has opened opportunities to extend this investigation, which has been conducted through this study.

Thus, the main goal of this work was to study the influence of the type of material of the balls on the dragging process of the particles and consequent wear mechanisms. For this, two abrasives were used, alumina and diamond, with similar particle sizes. TiN coatings were chosen because they have already been widely characterized by researchers, being commonly used, for example, in cutting tools for machining where the phenomenon of abrasion is quite significant. Based on the observations made, it was possible to conclude the following:It was found that the SS AISI 304 ball is undoubtably more suitable for ball-cratering micro-abrasion tests, as its material is slightly softer than the one usually used, AISI 52100, allowing to drag more particles and promoting a more uniform wear in the samples;The craters generated by SS AISI 304 balls are larger, allowing a better resolution between tests made with a different number of rotation cycles of the ball, or maybe reducing the duration of each test, but producing similar results to harder balls used in this type of test;Regarding the tracks produced on the balls’ surface, it was found that AISI 52100, as previously reported, presents high tendency to produce a deep and clear central groove, while SS AISI 304 balls tend to generate a more homogeneous track. This is related to the hardness presented by each one of the materials used in the balls. It must be referred to that AISI 52100 steel balls are in a heat-treated state, ready to use in ball-bearing;Regarding the evolution of the roughness, it was found that average roughness Ra increased in AISI 52100 balls (track area), presenting the main central groove in the middle of the track, while the tracks on SS AISI 304 balls become smoother in terms of Ra, but rougher in terms of Sz. This corroborates the lower hardness presented by SS AISI 304 balls, being deeply and abundantly scratched by small abrasive particles, but the rolling wear being the predominant wear mode, and due to its lower hardness, the tracks on SS AISI 304 balls are smoother, but present higher Sz due to deep scratches present on their surface;Moreover, it was found that even when using dissimilar abrasive particles in the ball-cratering tests, the wear phenomena on the balls’ surface are the same for each one of the materials used in the balls;Furthermore, although a mixed wear mode has been found in most of the tests, the rolling wear was predominant when using SS AISI 304, while grooving is predominant for AISI 52100 steel. Moreover, larger particles always tend to bypass the contact using the surrounding area and not contribute in a more significant way to carry the load in the contact area. This cannot be considered in an absolute way, but as a general trend;It was also found that the abrasive particles’ material is an important issue in this kind of test, but the main factor conditioning the results is the ball material. However, diamond particles have shown to be more suitable for micro-abrasion due to the well-defined craters generated, opposite to the ill-defined craters generated by alumina. This fact was already previously reported by other authors [16];The tests have also shown that the PTFE balls proved to be unsuitable for this type of test because they are self-lubricated and because they cannot capture, in an effective way, a large number of particles. In addition, the punctual charge ends up causing a deformation in the ball during the contact of the PTFE against the sample due to its lack of resistance, resulting in very irregular craters.

Thus, the outcomes of this work represent a real extension of the results and knowledge previously reported by other authors.

## Figures and Tables

**Figure 1 materials-14-00668-f001:**
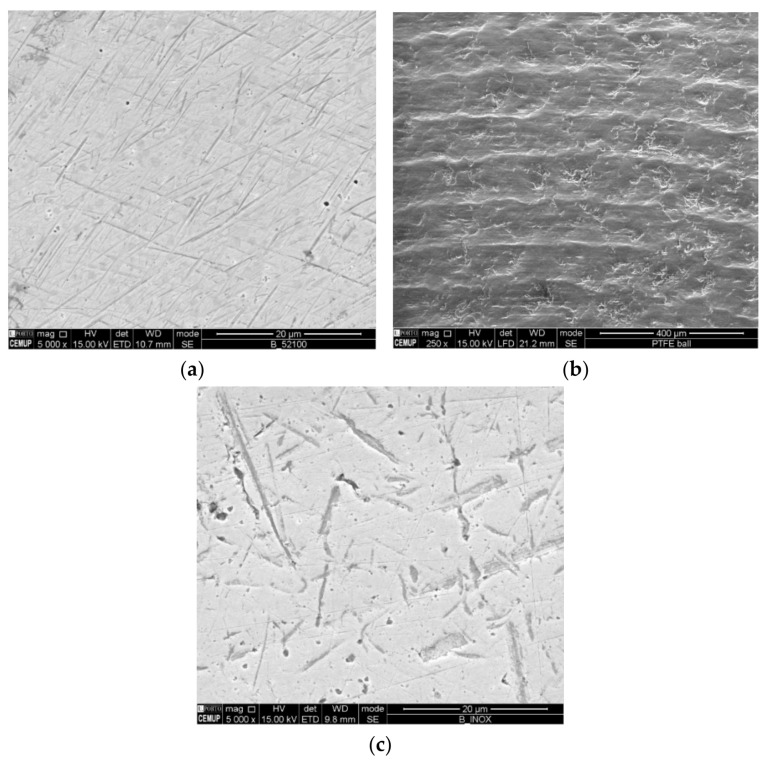
Balls’ morphology before testing: (**a**) AISI 52100 steel balls polished state; (**b**) Teflon PTFE balls; (**c**) SS AISI 304 steel balls polished state.

**Figure 2 materials-14-00668-f002:**
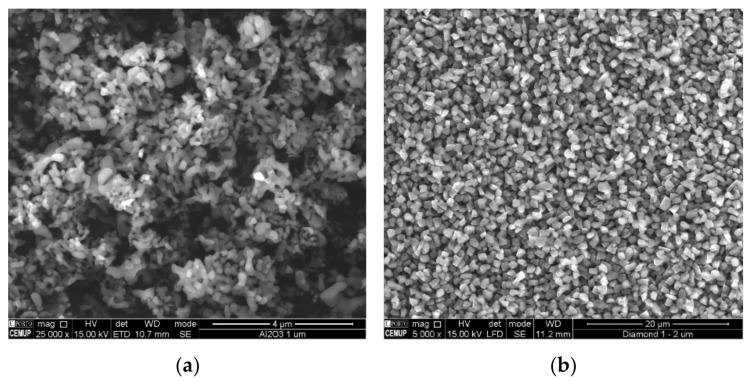
Morphological characterization of the abrasive particles geometry used: (**a**) alumina powder 1 µm and (**b**) diamond powder 1–2 µm.

**Figure 3 materials-14-00668-f003:**
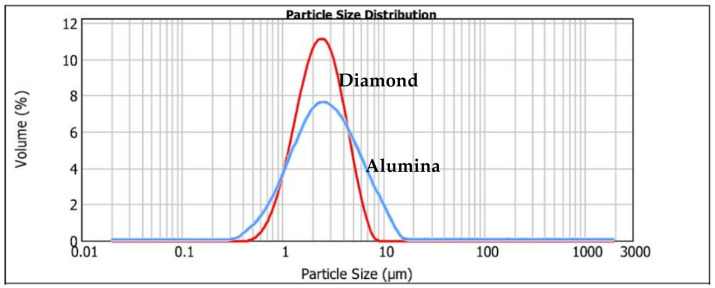
Distribution of the particles’ size regarding each type of abrasive particle: alumina 1 µm, and diamond powder 1–2 µm.

**Figure 4 materials-14-00668-f004:**
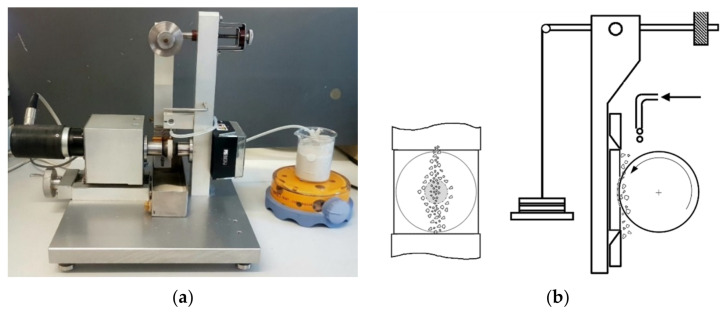
(**a**) Ball-cratering tribometer equipment used in this work. (**b**) Schematic diagram of the preferential path assumed by abrasive particles provided with different sizes within the range announced by the powder providers.

**Figure 5 materials-14-00668-f005:**
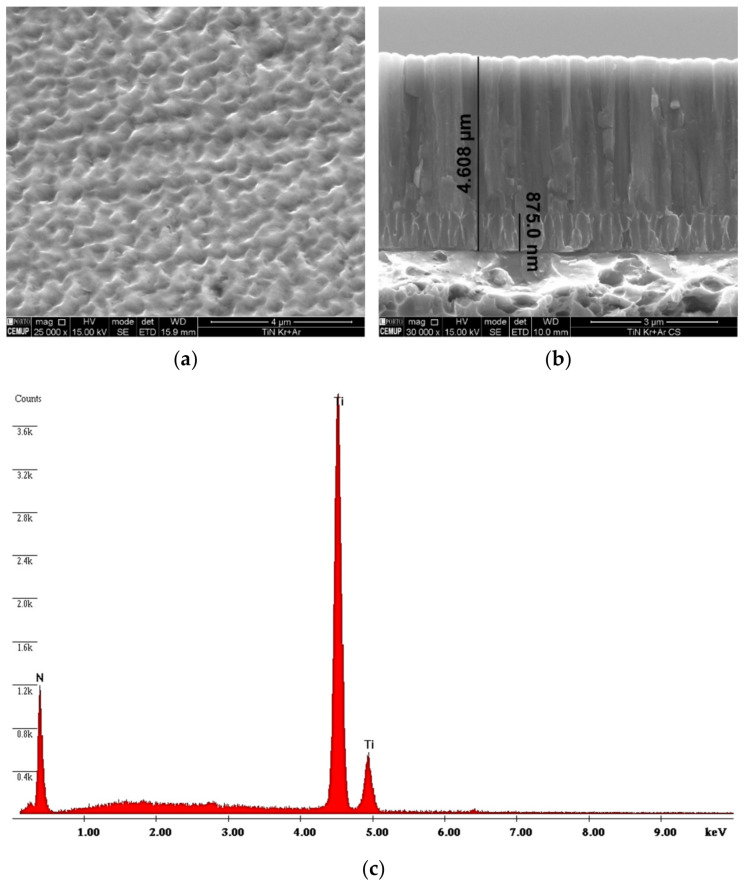
Morphological characterization of the film: (**a**) surface, (**b**) cross-section, (**c**) EDS spectrum.

**Figure 6 materials-14-00668-f006:**
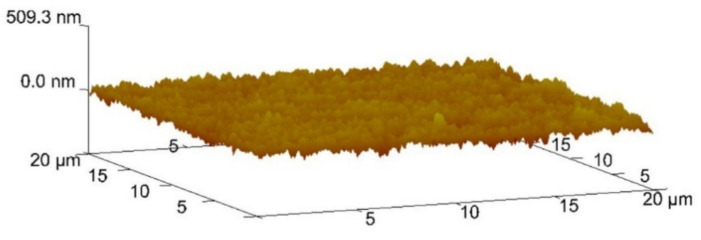
AFM topography analysis on the coated surface.

**Figure 7 materials-14-00668-f007:**
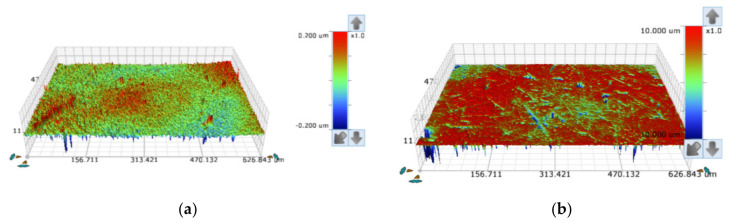
Optical profilometry topography analysis on the balls: (**a**) AISI 52100 steel balls polished state; (**b**) SS AISI 304 steel balls polished state.

**Figure 8 materials-14-00668-f008:**
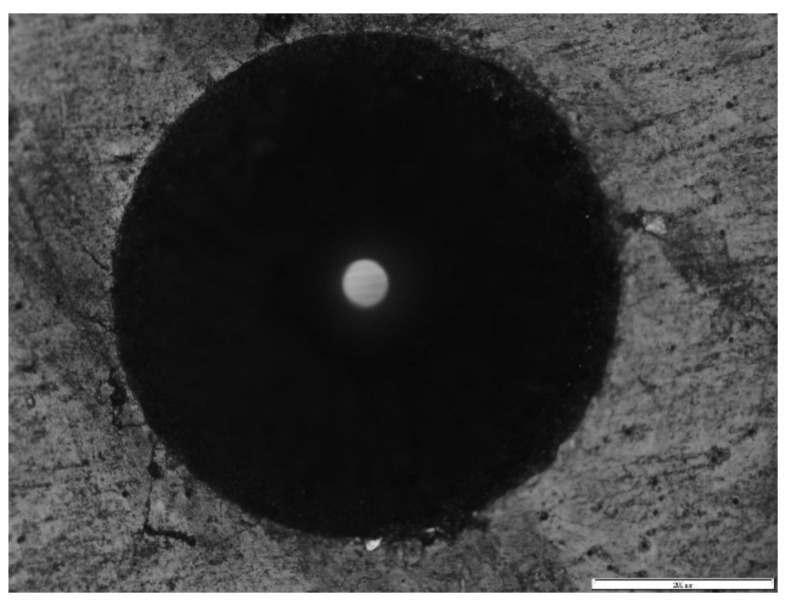
Optical microscope images of the indentation carried out on the coating’s surface.

**Figure 9 materials-14-00668-f009:**
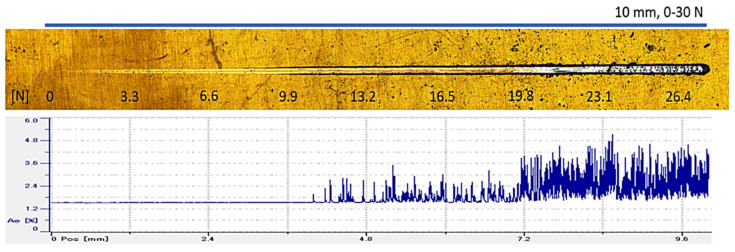
Measured width of the scratch (0–10 mm) and scratch test progressive load (0–30 N).

**Figure 10 materials-14-00668-f010:**
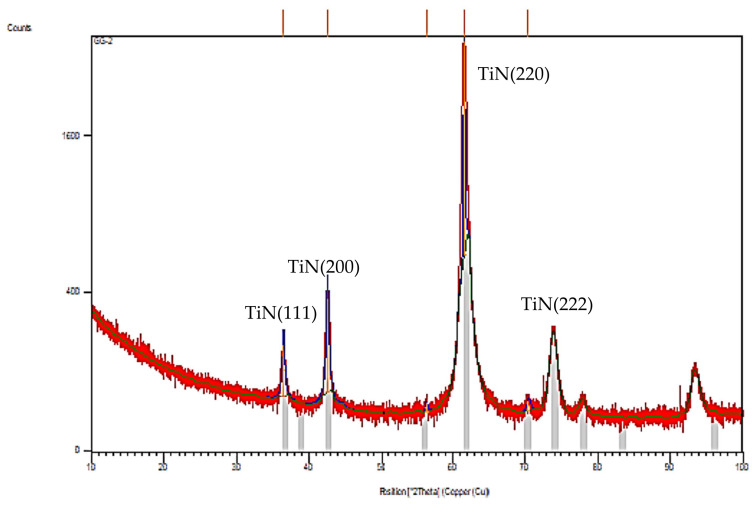
X-ray diffraction pattern of TiN coatings used in this work.

**Figure 11 materials-14-00668-f011:**
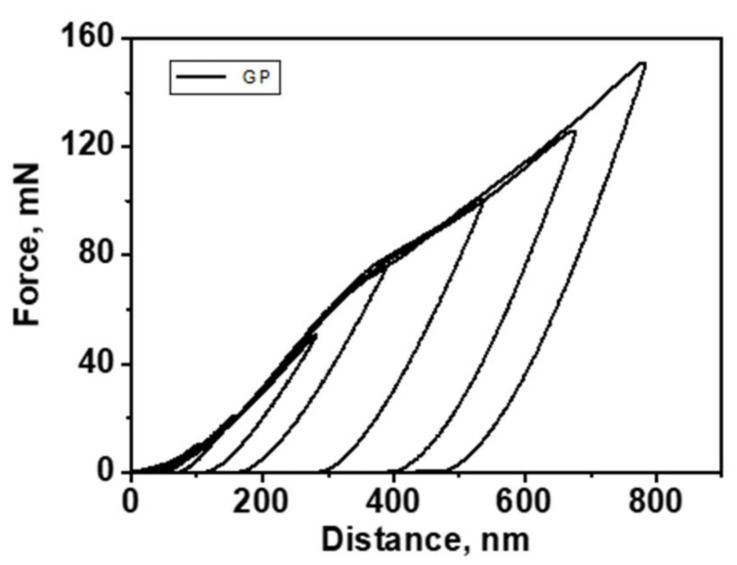
Load–discharge curve, resulting from a micro-hardness test.

**Figure 12 materials-14-00668-f012:**
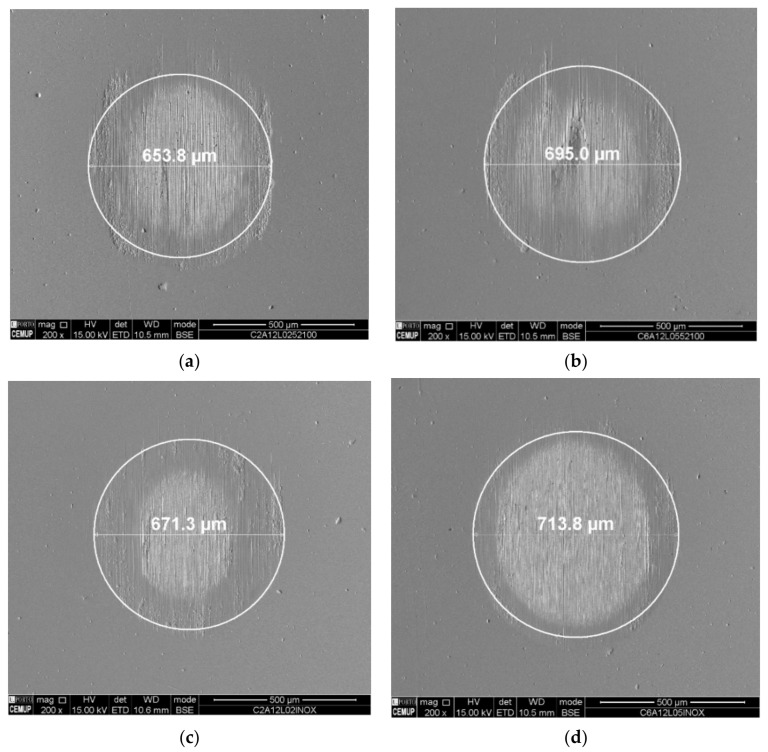
Wear craters generated on the test using alumina powder (1 µm) as abrasive: (**a**) AISI 52100; load 0.2 N, (**b**) AISI 52100; load 0.5 N, (**c**) SS AISI 304; load 0.2 N, (**d**) SS AISI 304; load 0.5 N.

**Figure 13 materials-14-00668-f013:**
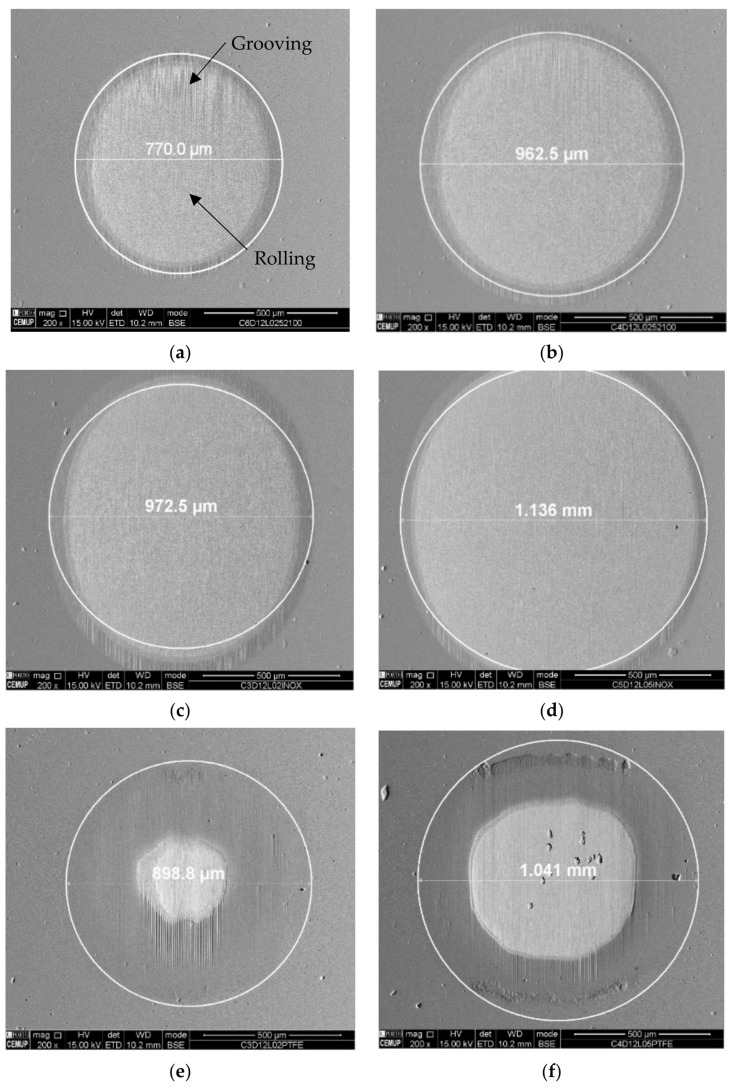
Wear craters generated on tests performed with diamond powder (1–2 µm) abrasive particles: (**a**) AISI 52100; load 0.2 N, (**b**) AISI 52100; load 0.5 N, (**c**) SS AISI 304; load 0.2 N, (**d**) SS AISI 304; load 0.5 N, (**e**) PTFE; load 0.2 N, (**f**) PTFE; load 0.5 N.

**Figure 14 materials-14-00668-f014:**
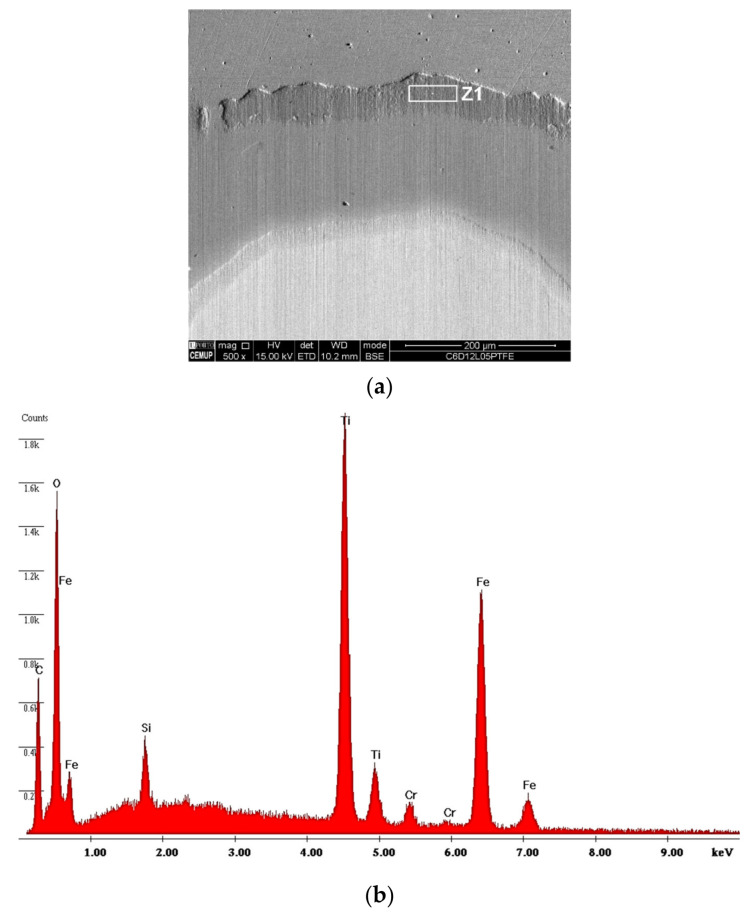
Detail of a wear crater generated by PTFE balls for diamond abrasive particles (1–2 µm) with a 0.5 N load: (**a**) crater inlet and (**b**) corresponding EDS spectrum in the Z1 zone indicating the presence of TiN.

**Figure 15 materials-14-00668-f015:**
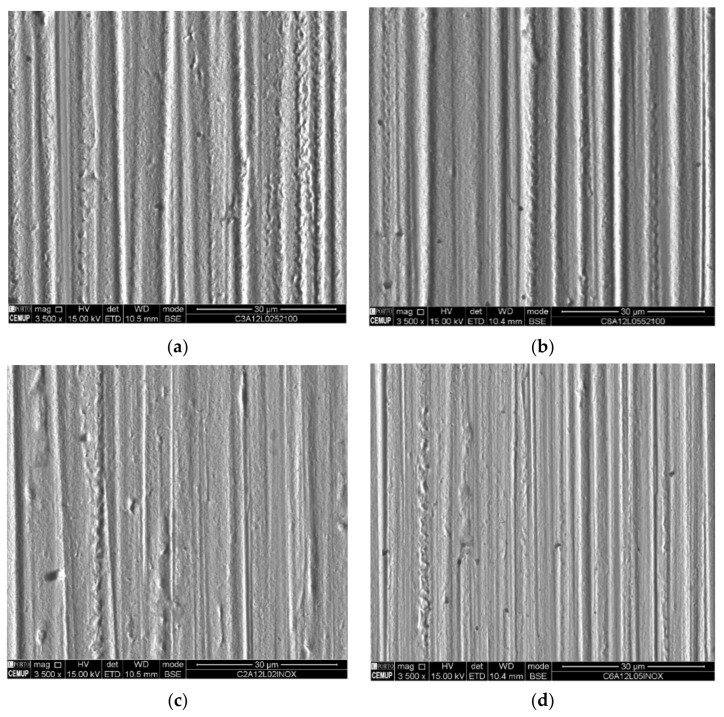
Evolution of the groove marks visible on the craters by SEM for the alumina (1 µm) abrasive: (**a**) AISI 52100; load 0.2 N, (**b**) AISI 52100; load 0.5 N, (**c**) SS AISI 304; load 0.2 N, (**d**) SS AISI 304; load 0.5 N.

**Figure 16 materials-14-00668-f016:**
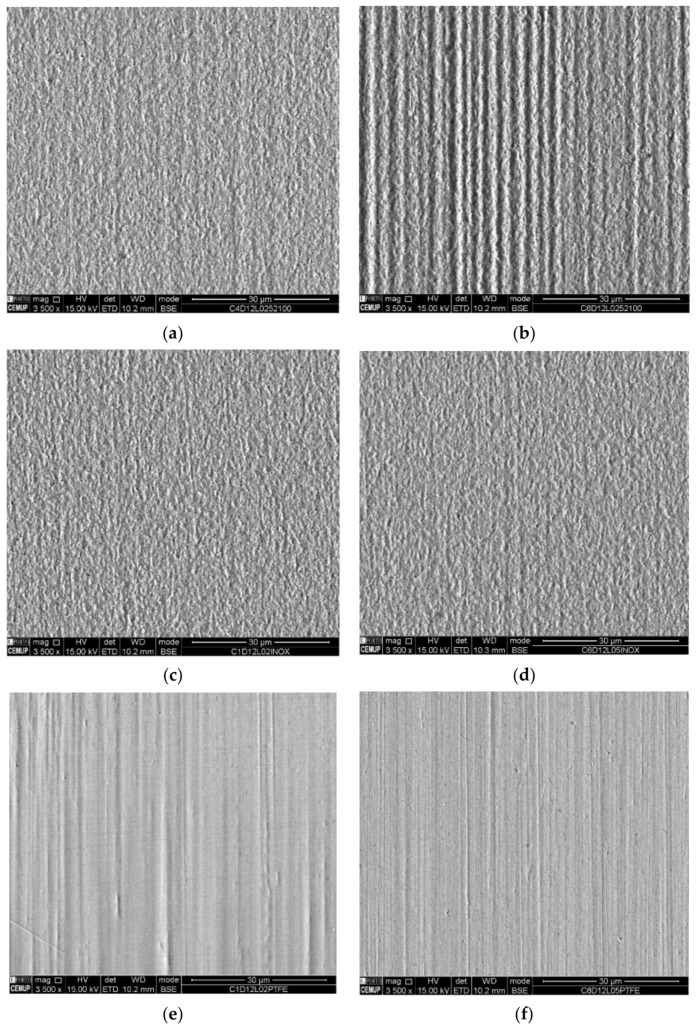
Evolution of the groove marks visible on the craters by SEM for the diamond (1–2 µm) abrasive: (**a**) AISI 52100; load 0.2 N, (**b**) AISI 52100; load 0.5 N, (**c**) SS AISI 304; load 0.2 N, (**d**) SS AISI 304; load 0.5 N, (**e**) PTFE; load 0.2 N, (**f**) PTFE; load 0.5 N.

**Figure 17 materials-14-00668-f017:**
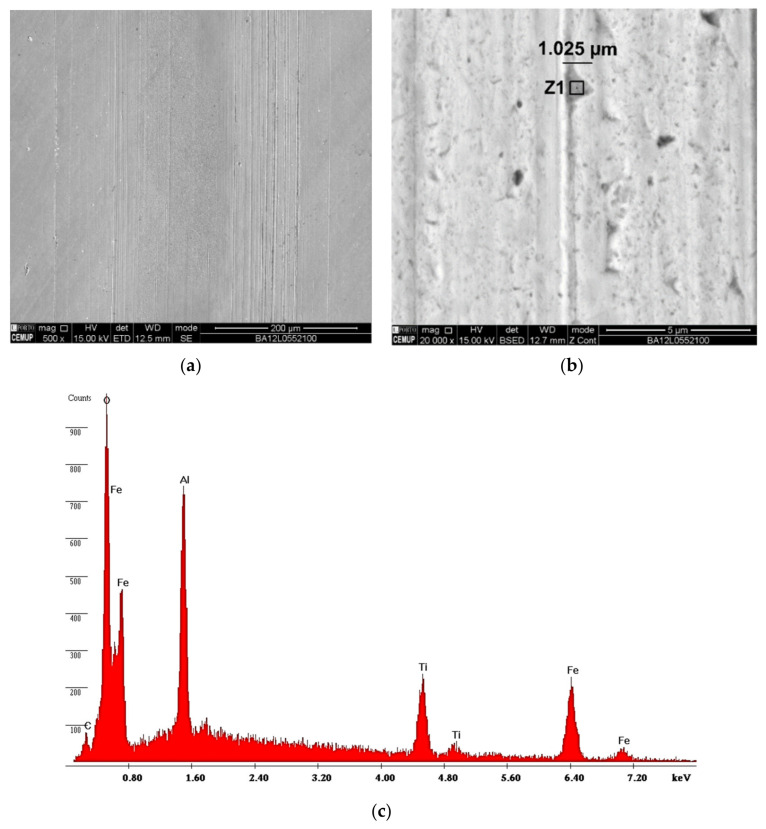
SEM track analysis for abrasive alumina powder (1 µm) with 0.5 N load for AISI 52100 ball: (**a**) track on the ball, (**b**) embedded alumina particle, (**c**) EDS spectrum in the Z1 zone indicating alumina particle.

**Figure 18 materials-14-00668-f018:**
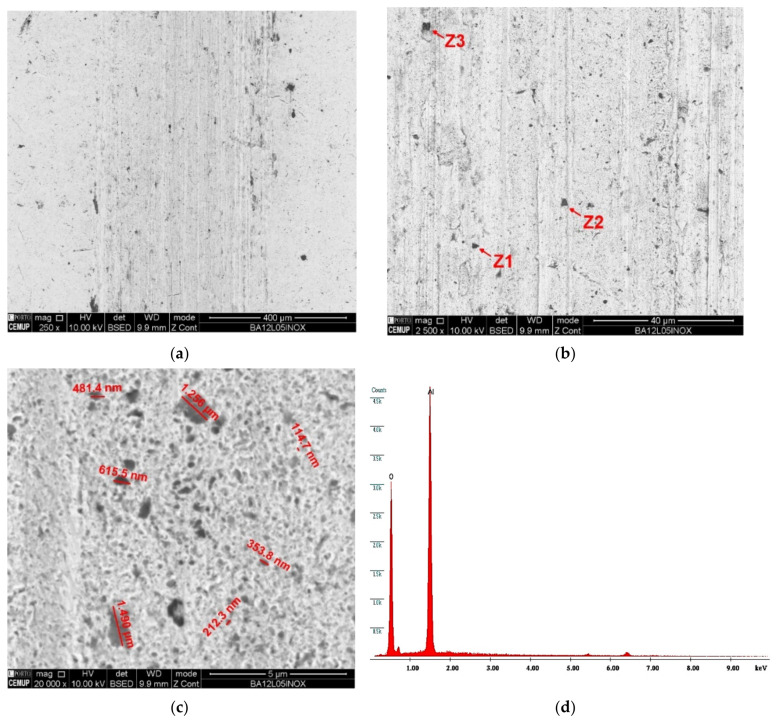
SEM track analysis for abrasive alumina powder (1 µm) with 0.5 N load for SS AISI 304 ball: (**a**) track on the ball, (**b**) embedded alumina particle, (**c**) alumina particle dimension, (**d**) EDS spectrum in the Z2 zone indicating alumina particle.

**Figure 19 materials-14-00668-f019:**
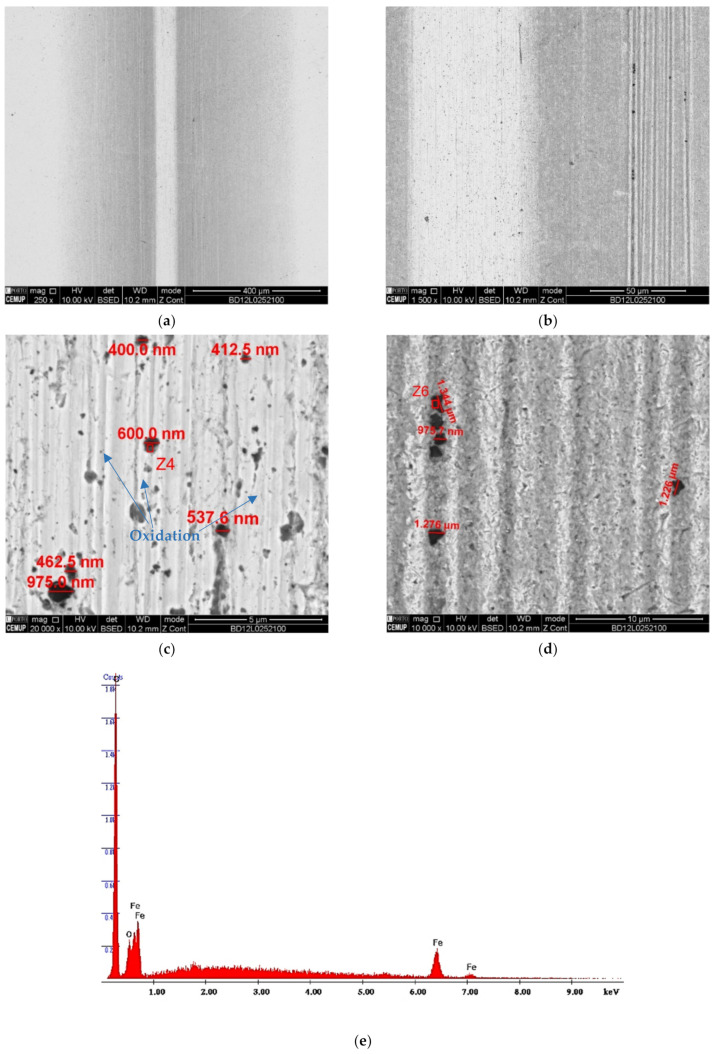
SEM track analysis for abrasive diamond powder (1–2 µm) with 0.2 N load for AISI 52100 ball: (**a**) track on the ball, (**b**) track on the ball detail analyzed, (**c**) embedded diamond particle dimension on the central mark, (**d**) embedded diamond particle dimension on the sides, (**e**) EDS spectrum in the Z4 zone indicating diamond particle, (**f**) EDS spectrum in the Z6 zone indicating diamond particle.

**Figure 20 materials-14-00668-f020:**
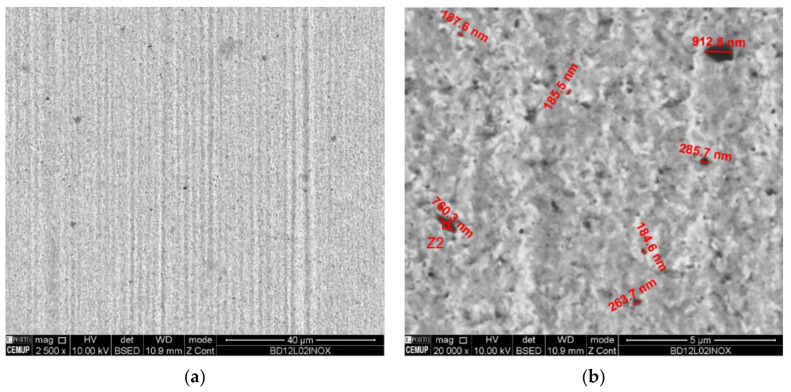
SEM track analysis for abrasive diamond powder (1–2 µm) with 0.2 N load for SS AISI 304 ball: (**a**) track on the ball, (**b**) embedded diamond particle dimension, (**c**) EDS spectrum in the Z2 zone indicating diamond particle.

**Figure 21 materials-14-00668-f021:**
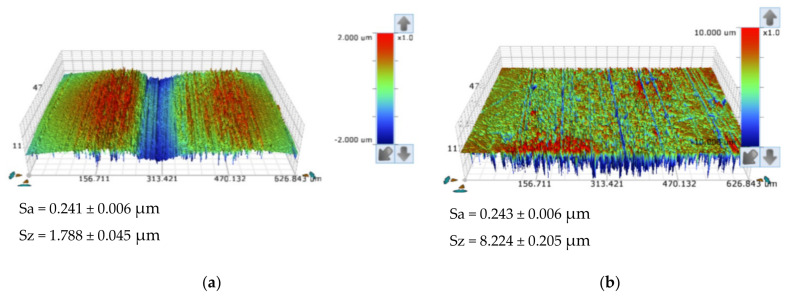
Representative images of the optical profilometry of the rail surface of the test balls for alumina powder (1 µm), load 0.5 N: (**a**) AISI 52100; (**b**) SS AISI 304.

**Figure 22 materials-14-00668-f022:**
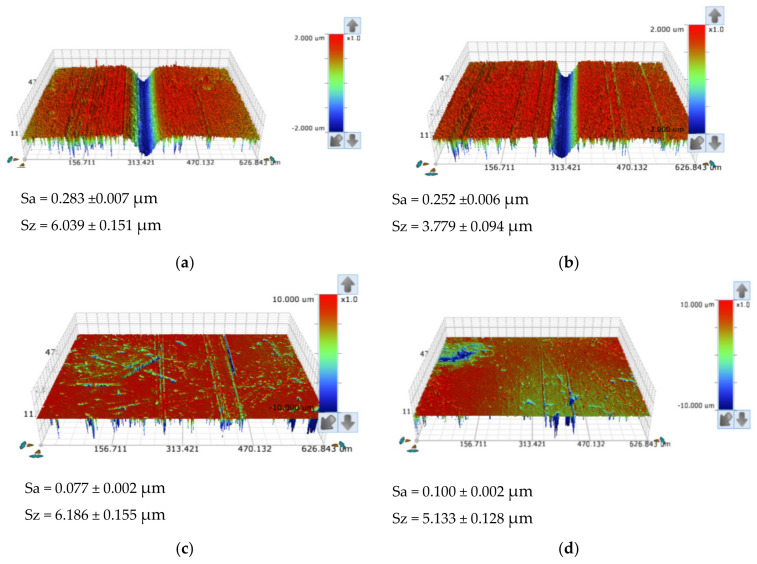
Representative images of the optical profilometry of the track surface of the test balls for diamond powder (1–2 µm): (**a**) AISI 52100; load 0.2 N, (**b**) AISI 52100; load 0.5 N, (**c**) SS AISI 304; load 0.2 N, (**d**) SS AISI 304; load 0.5 N.

**Table 1 materials-14-00668-t001:** Crater diameter according to the average size of the abrasive particles and ball material using 0.2 and 0.5 N as normal loads.

Abrasive	Load (N)	Ball Material	Ø Medium Outer (mm)	V—Volume Wear (mm^3^)	K—Wear Coefficient (mm^3^/Nm)
**Alumina Powder** **(1 µm)**	0.2	AISI 52100	0.671 ± 0.0251	0.00079 ± 0.00012	0.00010 ± 0.00002
		SS AISI 304	0.678 ± 0.0166	0.00082 ± 0.00008	0.00010 ± 0.00001
		PTFE	* N.A.	* N.A.	* N.A.
	0.5	AISI 52100	0.688 ± 0.0201	0.00087 ± 0.00010	0.00004 ± 0.000005
		SS AISI 304	0.745 ± 0.0241	0.00120 ± 0.00015	0.00006 ± 0.00001
		PTFE	* N.A.	* N.A.	* N.A.
**Diamond Powder** **(1–2 µm)**	0.2	AISI 52100	0.778 ± 0.0067	0.00142 ± 0.00005	0.00018 ± 0.00001
		SS AISI 304	0.941 ± 0.0404	0.00306 ± 0.00050	0.00038 ± 0.00006
		PTFE	0.892 ± 0.0052	0.00245 ± 0.00006	0.00031 ± 0.00001
	0.5	AISI 52100	0.963 ± 0.0051	0.00333 ± 0.00007	0.00017 ± 0.000004
		SS AISI 304	1.148 ± 0.0117	0.00673 ± 0.00027	0.00034 ± 0.00001
		PTFE	1.013 ± 0.0210	0.00409 ± 0.00034	0.00020 ± 0.00002

* N.A.—not available.

## Data Availability

Data sharing is not applicable to this article.

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
