# Peer review of "Study on the Influence of the Ball Material on Abrasive Particles’ Dynamics in Ball-Cratering Thin Coatings Wear Tests"

_materials, 2021, doi:10.3390/ma14030668_

Round 1

Reviewer 1 Report

The list of detailed remarks are given below:

  1. Please add the standard code of the substrate material (line 180).
  2. Fig. 2 - it will be better to compare images with the same magnification.
  3. Line 231 - there is an error, please correct it.
  4. XRD analysis methodology - what was the step and time for step?
  5. The dwell time in micro-harndess measurements is extreamly low, please explain it.
  6. The siliding distance is too short, please explain it. In my opinion the friction coefficient did not stabilize.
  7. Line 359 - such high accuracy? Please check it.
  8. Fig. 8 - there is load from 0 to 30 N, whereas Authors in Methodology chapter wrote, that load increased from 0 to 100 N, please explain.
  9. Instrumented indentation results - there is only value without standard deviation. Only one measurement has been carried out?
  10. The same in abrasion results (table 1), there is no standard deviation.

Author Response

Dear Reviewer #1,

The authors would like to thank the editor and the reviewers for taking the time to review this paper and provide positive feedback, useful suggestions, and valuable criticisms. We have carefully considered reviewer’s comments and believe the paper has been deeply improved.

Below you can find the questions put by the reviewers as well as our responses. We also presented a full revised version of the paper, where we try to include all the suggestions made by the reviewers.

Reviewer #1

Comment/Suggestion

Action done

The list of detailed remarks are given below:

Please add the standard code of the substrate material (line 180).

The authors are grateful for all the reviewer's comments. All guidelines had been followed. The authors added the information as suggested.

Fig. 2 - it will be better to compare images with the same magnification.

All comments are welcome, however, the images in figure 2 (a) and (c) have the same magnification corresponding to the steel balls used. The PTFE ball (b) has a polymer morphologically very different from steel balls, it is only possible to observe its texture in general with 250x, because a higher magnification will miss a wider view of the surface as can be seen in the figure below.

Line 231 - there is an error, please correct it.

We appreciate your comment. The sentence has been changed to “The obtained values for the analyzed abrasives are as follows”.

XRD analysis methodology - what was the step and time for step?

The authors thank you for your comment. The duration of the XRD trial lasted approximately 2 hours, Step size [2θ] = 0.0100 and scan step time: 1.000 s,

Method = Top of smoothed peak. This information has been added in the article.

The dwell time in micro-harndess measurements is extreamly low, please explain it.

Thank you for the comment. Indeed, it was a mistake. The dwell time has been changed to “30 s”.

The siliding distance is too short, please explain it. In my opinion the friction coefficient did not stabilize.

The authors appreciate the comment. It is a distance usually used by other researchers, who only performed 100 cycles and those who carried out 1000 cycles. In our study, 500 cycles have been used.

For this calculation: P = 25xPi = 78.54 mm; S = 78.54x500 = 39270 mm = 39.27 m. Reference 23 supports the statement, the authors studied the following sliding distances: 11.78 m, 29.44 m, 58.88 m, 117.75 m and 235.50 m. This is just an example.

Line 359 - such high accuracy? Please check it.

We appreciate the comment, however, the results obtained were in nanometers, for example for Rz = 0.029 ± 0.002 µm = 29 ± 2nm about 14% uncertainty. The values shown are correct.

Fig. 8 - there is load from 0 to 30 N, whereas Authors in Methodology chapter wrote, that load increased from 0 to 100 N, please explain.

Thank you for the comment. Indeed, it was a mistake.  Indeed, the value pointed out in the methodology is usually used in our group, but in this work we used a lower value, which was selected after some preliminary tests.

Instrumented indentation results - there is only value without standard deviation. Only one measurement has been carried out?

Indeed, the authors had not made this reference. The following sentence has been added: “In order to improve the accuracy of the measurement, ten indentations have been made.”

The same in abrasion results (table 1), there is no standard deviation.

Table 1 refers the deviations ( ) mentioned by the reviewer. The table has been improved to make this information clearer.

All manuscript was deeply revised in terms of misspellings and coherence, trying to avoid mistakes.

The authors would like to thank once again all the valuable contributions given by the Editor and Reviewers, allowing the paper improvement. We are looking forward to hearing from you. Thank you so much for your attention.

Kind regards,

Francisco Silva

Reviewer 2 Report

Recommendations for authors:

  1. Authors must indicate the usefulness of the new knowledge generated by the research
  2. Authors must add a diagram and view of the test stand
  3. The authors must remove the Spanish descriptions from Fig. 3
  4. In Table 1, the Authors must add the uncertainty of determination or measurement of individual quantities
  5. For a better understanding, the authors must add a diagram of the ball-particle cooperation
  6. The authors must clearly indicate the traces of damage in the SEM photos and identify the damage mechanisms caused by the paste particles
  7. The paper lacks a clear explanation of one reason for the work "... in order to understand the abrasive particles dynamics".

Author Response

Dear Reviewer #2,

The authors would like to thank the editor and the reviewers for taking the time to review this paper and provide positive feedback, useful suggestions, and valuable criticisms. We have carefully considered reviewer’s comments and believe the paper has been deeply improved.

Below you can find the questions put by the reviewers as well as our responses. We also presented a full revised version of the paper, where we try to include all the suggestions made by the reviewers.

Reviewer #2

Comment/Suggestion

Action done

Recommendations for authors:

Authors must indicate the usefulness of the new knowledge generated by the research

The authors are grateful for all the reviewer's comments. All guidelines had been followed. As pointed out in the manuscript, a similar study was done in the recent past [17]. Our study intends to extend the knowledge about the same topic. This is clearly referred on the manuscript /4th line of the Abstract + Line 24 (Abstract) + Line 620.

Authors must add a diagram and view of the test stand

Thank you for your comment. The authors added two bench images from the micro-abrasion tests, figure 4 (a).

The authors must remove the Spanish descriptions from Fig. 3

Thank you for your comment. Indeed, the authors made this mistake and made the respective correction.

In Table 1, the Authors must add the uncertainty of determination or measurement of individual quantities

The authors appreciate the comment. The table has been improved to make clearer this information.

For a better understanding, the authors must add a diagram of the ball-particle cooperation

As mentioned in point 2, a deposition of the abrasive on the rotating ball image has been added figure 4 (b).

The authors must clearly indicate the traces of damage in the SEM photos and identify the damage mechanisms caused by the paste particles.

Thank you for your comment. The information you are requesting is properly described on the text. This has been improved after the reference of Figure 11 on the text, as well as over Figure 12(a). Hope now this is perfectly clear.

The paper lacks a clear explanation of one reason for the work "... in order to understand the abrasive particles dynamics".

As pointed out in the manuscript, a similar study was done in the recent past [17]. Our study intends to extend the knowledge about the same topic. This is clearly referred on the manuscript /4th line of the Abstract + Line 24 (Abstract) + Line 620.

All manuscript was deeply revised in terms of misspellings and coherence, trying to avoid mistakes.

The authors would like to thank once again all the valuable contributions given by the Editor and Reviewers, allowing the paper improvement. We are looking forward to hearing from you. Thank you so much for your attention.

Kind regards,

Francisco Silva

Reviewer 3 Report

The paper is interesting and presents a detailed investigation of abrasive particle dynamics for testing of thin coatings. The study was conducted in an orderly and structured manner.

One of the problems is that the authors conclude about the rolling/dragging behavior of the particles based solely on the appearance of the wear track (e.g. lines 604, 647), which is an indirect not necessarily reliable observation. This should be mentioned in the manuscript.

Otherwise, mostly minor issues were observed, which are handwritten into the scanned hard-copy of the manuscript attached to this review report. It is suggested the authors address the suggested corrections and provide the corrections accordingly.

Author Response

Dear Reviewer #3,

The authors would like to thank the editor and the reviewers for taking the time to review this paper and provide positive feedback, useful suggestions, and valuable criticisms. We have carefully considered reviewer’s comments and believe the paper has been deeply improved.

Below you can find the questions put by the reviewers as well as our responses. We also presented a full revised version of the paper, where we try to include all the suggestions made by the reviewers.

Reviewer #3

Comment/Suggestion

Action done

The paper is interesting and presents a detailed investigation of abrasive particle dynamics for testing of thin coatings. The study was conducted in an orderly and structured manner.

One of the problems is that the authors conclude about the rolling/dragging behavior of the particles based solely on the appearance of the wear track (e.g. lines 604, 647), which is an indirect not necessarily reliable observation. This should be mentioned in the manuscript.

Otherwise, mostly minor issues were observed, which are handwritten into the scanned hard-copy of the manuscript attached to this review report. It is suggested the authors address the suggested corrections and provide the corrections accordingly.

The authors are grateful for all the reviewer's comments. All guidelines had been followed, which greatly increased the quality of the work. Indeed, you are right, but this is the way usually used by Researchers to study this kind of wear mechanisms. Thus, some text has been added as requested by you, highlighting this detail.

Line 186: how was the Ra measured?

The authors appreciate the comment. These values were measured using a profilometer. Thus, the following text has been added: “…using a Mahr Perthometer M2 and carrying out five measurements in two orthogonal directions, and calculating the average value and corresponding standard deviation.”

Line 219: define what span is?

This information was added in the text.

Line 364: on which basics was this concluded?

Thank you for your comment. A short explanation has been added to the text and can be seen before Figure 6.

Line 371: this is correct?

Thank you for your question. Indeed, the value is correct, but the parameter was wrong (Sz, not Rz), because the analysis was made in 3D.

Line 426: of what?

Thank you for your pertinent question. Indeed, the idea was incomplete, but we added the corresponding explanation after Figure 11 (highlighted in yellow colour as usual).

Line 459: for comparison SEM images could be presented anyway.

Line 471: what about the wear craters created by the PTFE ball?

We understand the Reviewer’s concern about this issue. However, we have not images, because after an exhaustive search to find them, it was not possible to identify some mark of the contact between the ball and the sample. In fact, because the hardness of the PTFE ball is too different from the sample and abrasive particles’ hardness, perhaps the ball deforms under the normal load and the abrasive particle doesn’t produce abrasive wear. This is mainly observed when using alumina abrasive particles. However, with diamond particles, small craters have been observed, as shown in Figure 13 (e, f).

Line 522: roughness of the balls?

Yes, it is correct, the authors refer to the roughness of SS AISI 304 balls.

Line 525: wear phenomenon is rolling.

Thank you so much for your pertinent and accurate comment. Indeed, it was our mistake. The wear phenomenon is clearly grooving, not rolling, as can be seen in Figure 16 (e,f). The mistake has been corrected.

Line 345

Thank you so much for your comment. However, what it is possible to see are two kind of dark areas: the darkest areas are abrasive particles, and the less dark are oxidation marks. Thus, we felt the need to add a text box in Figure 19 (c) to explain this fact.

Line 599

Thank you so much for expressing your concern in terms of consistency. However, after a deep search, we didn’t find any inconsistence regarding the material of the balls. For us, it is clear that SS AISI304 presents better dragging effect of the abrasive particles.

All manuscript was deeply revised in terms of misspellings and coherence, trying to avoid mistakes.

The authors would like to thank once again all the valuable contributions given by the Editor and Reviewers, allowing the paper improvement. We are looking forward to hearing from you. Thank you so much for your attention.

Kind regards,

Francisco Silva

Round 2

Reviewer 1 Report

All my remarks have been included.

Only one issue - Table 1, please standardize the accuracy for average and standard deviation values.

Reviewer 2 Report

Thank you for the changes and explanations provided. I have no further comments on your work. Respectfully